# Evaluation of the Effects of Genistein In Vitro as a Chemopreventive Agent for Colorectal Cancer—Strategy to Improve Its Efficiency When Administered Orally

**DOI:** 10.3390/molecules27207042

**Published:** 2022-10-19

**Authors:** Juan Pablo Rendón, Ana Isabel Cañas, Elizabeth Correa, Vanesa Bedoya-Betancur, Marlon Osorio, Cristina Castro, Tonny W. Naranjo

**Affiliations:** 1Medical and Experimental Mycology Group, CIB-UPB-UdeA-UDES, Corporación para Investigaciones Biológicas, Carrera 72 A # 78B-141, Medellin 050034, Colombia; 2School of Engineering, Universidad Pontificia Bolivariana, Circular 1 # 70-01, Medellin 050031, Colombia; 3School of Health Sciences, Universidad Pontificia Bolivariana, Calle 78 B # 72 A-109, Medellin 050034, Colombia

**Keywords:** genistein, encapsulation, colon cancer, chemoprevention

## Abstract

Colorectal Cancer (CRC) ranks third in terms of incidence and second in terms of mortality and prevalence worldwide. In relation to chemotherapy treatment, the most used drug is 5-fluorouracil (5-FU); however, the use of this drug generates various toxic effects at the systemic level. For this reason, new therapeutic strategies are currently being sought that can be used as neoadjuvant or adjuvant treatments. Recent research has shown that natural compounds, such as genistein, have chemotherapeutic and anticancer effects, but the mechanisms of action of genistein and its molecular targets in human colon cells have not been fully elucidated. The results reported in relation to non-malignant cell lines are also unclear, which does not allow evidence of the selectivity that this compound may have. Therefore, in this work, genistein was evaluated in vitro in both cancer cell lines SW480 and SW620 and in the non-malignant cell line HaCaT. The results obtained show that genistein has selectivity for the SW480 and SW620 cell lines. In addition, it inhibits cell viability and has an antiproliferative effect in a dose-dependent manner. Increased production of reactive oxygen species (ROS) was also found, suggesting an association with the cell death process through various mechanisms. Finally, the encapsulation strategy that was proposed made it possible to demonstrate that bacterial nanocellulose (BNC) is capable of protecting genistein from the acidic conditions of gastric fluid and also allows the release of the compound in the colonic fluid. This would allow genistein to act locally in the mucosa of the colon where the first stages of CRC occur.

## 1. Introduction

According to the Global Cancer Observatory (GCO), in 2020, colorectal cancer (CRC) ranked as the type of cancer with the fourth highest incidence worldwide (approximately 19.5%). In women, an incidence rate of 23.4% is estimated and in men a rate of 16.2%. This type of cancer represents the third most common cause of death and is the second most prevalent for men and women of all ages worldwide [1]. Currently, examinations through colonoscopy have successfully detected early CRC, which makes it possible to search for different therapies to treat this pathology; however, 25% of patients are diagnosed with metastatic disease. Nonetheless, thanks to advances in medicine, the survival of these patients can be improved to more than two years with the combination of chemotherapy and biological agents [2].

In the clinical field, most cases of CRC are diagnosed as a terminal chronic condition or in a metastatic state due to the non-obvious development of the disease and symptoms during its initial phase. Additionally, the inevitable drug tolerance and side effects associated with chemotherapy make it more difficult to treat CRC effectively. Consequently, any candidate component with potential application for the treatment of metastatic CRC should be explored in depth [3]. In general, the choice for the treatment of CRC depends on several factors such as the clinical and health conditions of the patient, the size of the tumor, its location, and the presence of metastases. However, surgery remains the most common treatment option, especially for localized lesions [4]. Chemoradiation is sometimes required for locally advanced rectal cancer after surgical removal. Immunotherapy is also an option for metastatic CRCs that are microsatellite unstable [5]. When surgery is not necessary, treatment may include radiofrequency ablation, cryosurgery, chemotherapy, radiation therapy, or targeted therapy. For these cases, chemotherapy treatment is the most common and consists of the use of drugs that hinder tumor growth through the destruction of cancer cells, but the toxicity of chemotherapy increases with the age of the patients [6]. The drug most used in the chemotherapy of various types of cancer is 5-fluorouracil (5-FU), and it has been used with great success in CRC [7].

5-FU is an analog of uracil with a fluorine atom at the C-5 position instead of hydrogen, allowing it to rapidly enter the cell using the same facilitated transport mechanism as uracil. The mechanism of 5-FU cytotoxicity has been attributed to the misincorporation of fluorinated nucleotides into RNA and DNA and the inhibition of the nucleotide-synthesizing enzyme thymidylate synthase (TS) [7]. This drug can be given by continuous pump, 48 h infusion, weekly injections, or daily injections [2]. More than 80% of administered 5-FU is primarily catabolized in the liver and has shown toxic effects such as myelosuppression and other adverse gastrointestinal, hematologic, neural, and dermatologic side effects. Therefore, new therapeutic strategies are currently being sought to treat CRC that have fewer toxic effects at the systemic level [8].

Recent research has shown that various natural compounds have chemotherapeutic and anticancer effects; these investigations focus on the relationship of these effects with the specific biological targets associated with cancer on which these compounds act [9]. Among these compounds are flavonoids, which are a class of natural polyphenolic compounds present in vegetables, fruits, and soybeans. These compounds have been studied extensively in recent years in an attempt to understand the specific proteins on which they act to exert their anticancer functions. These functions and mechanisms have been evaluated in both in vitro and in vivo studies showing that flavonoids suppress carcinogenesis in various models of cancer cells, acting on multiple pathways involved in cell metabolism, apoptosis, adhesion, migration, and angiogenesis, as well as the immune response [10].

Genistein is a biologically active flavonoid found in high amounts in soybeans. Many studies have described a relationship between a soy-rich diet and cancer prevention, further demonstrating the pharmacological effects of genistein, including antiestrogenic action, antioxidant action, inhibition of angiogenesis, and anticancer activity against breast and ovarian cancer tumor cells. Therefore, it is considered as a promising chemopreventive agent in the treatment of cancer [3,11]. In this sense, chemoprevention is defined as a blockage, delay or reversal of a carcinogenic process through chemical and/or natural agents. Clinically, chemoprevention is classified as primary, secondary, or tertiary. Primary chemoprevention applies to the general population and to those who may be at risk of developing the disease. Secondary chemoprevention applies to patients with premalignant lesions that may progress to invasive disease. Tertiary chemoprevention is aimed at preventing disease recurrence in those who have already undergone potentially curative therapy. At the molecular level, cancer chemoprevention is characterized by the interruption, or at least the delay, of multiple pathways and processes in any of the three stages of carcinogenesis: initiation, promotion, and progression [12].

In CRC, several in vitro studies have shown that genistein exhibits growth-inhibitory activity and promotes apoptosis in a dose-dependent manner. Genistein causes cell cycle arrest in colon cancer cell lines HCT-116 and SW480, mainly participating in cell cycle regulation and apoptosis [13]. In other studies with the HT-29 cell line, it was observed that genistein inhibits EGF-induced proliferation, reverses the Epithelial–Mesenchymal Transition (EMT) and promotes the activation of apoptosis via caspase 3 [14,15]. Recently, genistein was shown to be able to inhibit cell invasion and migration of colon cancer cells by altering the expression of migration-associated factors and genes, including MMP9, MMP2, TIMP1, E-cadherin, β-catenin, c-Myc, and cyclin D1 [16]. Finally, other authors demonstrated that the treatment of HCT-116 cells with genistein causes inhibition of cell proliferation and induces apoptosis [17].

The following information demonstrates the ability of genistein to inhibit the proliferation and migration of tumor cells due to the inhibition of the activity of several molecular targets, making it a promising natural compound for the chemoprevention and treatment of CRC. However, the effect of this compound on non-malignant cell lines that show the selectivity of this compound has not yet been reported. By definition, an ideal compound should have a relatively high toxic concentration but a very low active concentration. Under this premise, the compound would affect cancer cells but should not affect non-malignant ones [18,19,20]. To date, there have been no reports of the evaluation of genistein on two cell lines derived from the same patient but with different stages of evolution of colorectal adenocarcinoma, and its role in other mechanisms of action such as necroptosis or its ability to regulate the expression of immunological markers in cells. Therefore, in this work, the effect of genistein on cancer cell lines and a non-malignant cell line was evaluated by means of in vitro tests. This would allow us not only to determine cell viability, but also to obtain the calculation of the selectivity index. Likewise, the effect that this natural compound exerts on antiproliferative and apoptotic activity, the production of reactive oxygen species (ROS) and the expression of immunological markers in these cell lines was evaluated as an approach strategy towards the possible mechanism of action involved.

Additionally, in this research, the encapsulation of genistein is proposed as a strategy to improve pharmacokinetic activity, achieving a better local effect at the target site of this compound when administered orally. This is because, despite the beneficial properties of genistein, the use of this compound in vivo is limited due to its low water solubility, rapid biotransformation to inactive metabolites, poor accumulation in target tissues and cells, and low concentration in the blood after oral administration [21,22].

## 2. Results

### 2.1. Encapsulation of Genistein

Figure 1 shows the empty BNC capsules (Figure 1a,b) and the BNC capsules loaded with genistein (Figure 1c,d), prepared by the spray-drying method. Empty BNC capsules have a particle size distribution between 1 and 5 µm, while BNC/GEN capsules have a particle size distribution between 1 and 6 µm, but in both cases, the particles are most often around 3 µm. In Figure 1b,d, it can be seen that the surfaces of the capsules are composed of a network of collapsed nanofibers in response to the evaporation of water and the establishment of irreversible hydrogen bonds. The ratio of genistein and BNC in the capsules was 5.52 mg GEN/1 g BNC and a spray drying yield of 52% was obtained.

### 2.2. In Vitro Release Assays in Gastrointestinal Fluids

To evaluate the maximum desorption capacity of genistein when it is encapsulated in BNC, a release of the compound was performed in simulated physiological fluids of the stomach, small intestine, and colon. In the release profiles found in Figure 2, it was observed that 8.7% of the compound is released in gastric fluid at 2 h, while 44.6% is released in small intestine fluid at 24 h. Finally, a genistein release of 92.5% in the colonic fluid was observed at 48 h.

For release into the stomach and intestine fluid, the experimental data had a better fit to a pseudo-first-order kinetic model with R^2^ = 0.968 and R^2^ = 0.888, respectively. In physiological stomach fluid, a gradual release of genistein is observed with a release rate of 0.001 min^−1^, and for the small intestine, a release rate of 0.021 min^−1^ was obtained, which stabilizes after 60 min. This is due to the protection generated by bacterial nanocellulose to the compound. On the other hand, the experimental data of the release in colonic fluid fit with R^2^ = 0.927 to a pseudo-second-order model. The release of genistein in this medium was much greater and in a controlled and prolonged manner.

### 2.3. Effect of Free Genistein on Cell Viability

Table 1 and Table 2 show the results obtained in the cell viability assay. The inhibitory concentration 50 (IC_50_) was determined by dose–response curves and the selectivity index by the formula IS = IC_50_ Control cells (HaCat)/IC_50_ Tumor cells (SW480 and SW620). A selectivity index greater than 1 indicates that the treatment is more cytotoxic to tumor cells than to control cells. The results show that genistein has selectivity in the SW480 and SW620 cell lines at 24 h and 48 h of treatment against the non-malignant HaCat cell line. Figure 3 shows that cell viability was inhibited in a dose-dependent manner in the three cell lines evaluated compared to the growth control.

Based on the IC50 value of free genistein, cytotoxicity assays of empty BNC capsules and BNC/GEN were performed. Figure 4 shows that the empty NCB capsules do not have a cytotoxic effect on any of the cell lines evaluated, since they showed a percentage of cell viability greater than 96% both at 24 and 48 h of incubation. Comparing the evaluated concentration of the IC50 of free and encapsulated genistein, it can be seen that there is a greater effect on the decrease in cell viability of BNC/GEN both at 24 and 48 h. This is associated with the protection offered by the BNC to the compound from the pH of the medium, which allows a controlled and prolonged release of it over time.

### 2.4. Antiproliferative Effect of Genistein

In Figure 5, it can be seen that genistein showed an antiproliferative effect throughout the evaluated times in a dose-dependent manner for both cell lines SW480 and SW620. In SW480 cells, genistein induced an inhibitory effect on viability from day 2 of treatment with high concentrations of the compound (37, 185, and 370 µM), while for days 4 and 6, this effect was observed at low concentrations (from 18.5 µM). Similar results were observed in the metastatic cell line (SW620), in which genistein was found to have a significant effect on cell viability on days 2 and 4 at all concentrations tested (3.7–379 µM). While for day 6, concentrations higher than 18.5 µM were required and for day 8, concentrations higher than 37 µM, concluding that the antiproliferative effect of genistein is conditioned by the dose and treatment time in these cell models. These results again demonstrate that there is a greater effect of genistein when the dose and exposure time to the compound are increased.

### 2.5. Production of Reactive Oxygen Species (ROS)

To evaluate ROS production in SW480 and SW620 cell lines, the IC_50_ value of genistein was used. Figure 6 shows that there is a greater production of ROS in the cells treated with genistein. In the SW480 cell line, a greater production of oxidative stress is observed at 24 h of treatment and in the SW620 cell line, this effect is observed both at 24 h and 48 h, showing in this last evaluation time higher levels of ROS, indicating that genistein can generate oxidative stress in this colorectal cancer cell line, possibly leading to the process of cell death.

### 2.6. Apoptotic Capacity of Genistein

The apoptotic capacity of genistein on cell lines SW480 and SW620 can be seen in Figure 7 and Figure 8. The results showed an increase in the population where DNA fragmentation was evidenced, with respect to the control in the treatments with genistein, both at 24 h and 48 h in both cell lines. By means of this assay, the APO-DIRECT^TM^ kit (Chemicon Cat. N°APT110) with PI/F-dUTP makes it possible to differentiate, at the top of each graph, the population where a cell death process takes place. For the SW480 cell line, higher percentages of cell death were observed compared to the control (77.8% and 21.9% at 24 h and 48 h, respectively). For the SW620 cell line, higher percentages of cell death were also observed compared to the control (44.2% and 30.3% at 24 h and 48 h, respectively).

To determine the intracellular proteins that participate in the apoptosis process of the SW480 and SW620 cell lines, Caspase 3, p53, Cytochrome c, BCL2, and cleaved PARP proteins were analyzed. In the SW620 cell line, after 24 h of treatment with genistein, a significant increase in Caspase 3 and cleaved PARP proteins was observed, and after 48 h of treatment, a significant increase in Caspase 3, p53, Cytochrome c, and cleaved PARP proteins was observed. The antiapoptotic protein BCL2 was not detected at 24 h and did not show significant differences compared to the control at 48 h of treatment (Figure 9). In the SW480 cell line, no levels of these proteins were detected at any of the two evaluation times with genistein (Results not shown).

### 2.7. Evaluation of Cytokine Expression

Cell supernatants were analyzed after 24 and 48 h incubation, with and without genistein. In the SW480 cell line, there were no significant changes in the cytokines tested (results not shown).

On the other hand, in the SW620 cell line, several cytokines showed a significant increase both at 24 h and 48 h after genistein treatment (Figure 10). At 24 h, a significant increase over the control was observed after treatment in the cytokines IL-1B, IL-2, IL-6, IL-13, IL-17A, IL-27, and GM-CSF. For 48 h of treatment, a significant increase in the cytokines IL-1B, IL-2, IL-4, IL-5, IL-10, IL-17A, IL-18, IL-27, and GM-CSF was observed. The other cytokines did not show significant differences from the control in this cell line (results not shown).

## 3. Materials and Methods

This work was carried out in the Laboratory of Medical and Experimental Mycology CIB-UdeA-UPB-UDES located in the Corporación para Investigaciones Biológicas, Medellín, Colombia.

A commercial genistein CAS No. 446-72-0 (Shanghai Yingrui Biopharma Co., Shanghai, China), with 98% purity, was used for the analyses.

### 3.1. Preparation of Bacterial Nanocellulose

Bacterial nanocellulose (BNC) was used as encapsulating agent, which was obtained through the fermentation of the bacterium *Komagataeibacter medellinensis* NBRC 3288, which was isolated in the Central Retail of Medellín and was identified in the Universidad Pontificia Bolivariana [23]. Nanocellulose was prepared using a modified Hestrin–Schramm (HS) culture medium with glucose at 2% (*w*/*v*), peptone at 0.5% (*w*/*v*), yeast at 0.5% (*w*/*v*), disodium phosphate at 0.267% (*w*/*v*), and citric acid to adjust the pH up to 3.5. After fermentation for 7 days, the BNC membranes were removed and purified in a 5% (*w*/*v*) KOH solution to remove biomass and debris. Finally, the NCB was processed in a monobloc blender and passed to a MKCA6-3 ultrafine friction mill (Masuko Sangyo^®^ Co., Ltd., Kawaguchi, Japan), with a total of 27 passes for the individualization of the nanofibers. The BNC was sterilized for later use.

### 3.2. Encapsulation of Genistein

The encapsulation of genistein in BNC was carried out using the spray-drying technique in a (BÜCHI Labortechnik AG, Flawil, Switzerland). First, a solution of 0.1% BNC (1 mg/mL) and 5.52 mg of genistein were prepared. Then, the solution was dried with a feed flow rate of 5 mL/min, an air flow of 35 m^3^/h, an air pressure of 6 bars, and an air inlet temperature of 150 °C.

### 3.3. Morphology and Size of Bacterial Nanocellulose/Genistein (BNC/GEN) Capsules

The morphology and size of the capsules obtained were analyzed by scanning electron microscopy (SEM) using JEOL JSM 6490 LV equipment (JEOL, Tokyo, Japan) in a high vacuum with a secondary electron detector to obtain high-resolution SEI images at an acceleration voltage of 20 kV. The capsules were placed on a carbon ribbon and coated with a thin layer of gold. The size of the capsules was measured using the free distribution software: Image J 1.49, adjusting each capsule to an ellipse. A total of 100 capsules were measured to find the size distribution.

### 3.4. In Vitro Release Assays in Gastrointestinal Fluids

For genistein release profiles, the membrane dialysis method with a size of 12–14 kDa was used [24] in simulated physiological fluids of the stomach, small intestine and colon with pH values of 1.2, 6.0 and 7.4, respectively, following the formulations proposed by Marques et al. [25]. These formulations allowed us to simulate the basal conditions of each organ taking into account the pH and osmolarity. The capsules inside the membranes were kept at 37 °C with a stirring speed of 80 rpm for 2 h for the stomach, 24 h for the intestine, and 48 h for the colon. At every time point, an aliquot of the fluid was taken and the total volume was replenished. Genistein concentration was determined by UV-Vis spectroscopy at 260 nm, using absolute ethanol as a solvent. Finally, the percentage and amount of genistein released from the capsules were fitted to pseudo-first- and second-order kinetic models [26].

### 3.5. Cell Lines

For the biological tests, the human colon adenocarcinoma cell line SW480, its metastatic derivative SW620, and the non-malignant cell line HaCaT were used. These cell lines were obtained from the European Collection of Authenticated Cell Cultures (ECACC, Salisbury, UK) and cultured in Dulbecco’s Modified Eagle Medium (DMEM), supplemented with 10% horse serum (Gibco, Waltham, Massachusetts, United States) heat-inactivated (60 °C), Penicillin/Streptomycin 1% (Sigma-Aldrich, Burlington, MA, USA) and Non-Essential Amino Acids 1% (Sigma-Aldrich). For the experiments, the serum concentration was reduced to 3% and the medium was supplemented with 5 mg/mL transferrin, 5 ng/mL selenium, and 10 mg/mL insulin (ITS Liquid Media Supplement 100×; Sigma-Aldrich) [27]. Before use, all cell lines were tested for the detection of *Mycoplasma* spp. by PCR, using specific primers [28].

### 3.6. Cell Viability Assay

Cell viability after genistein treatments was evaluated by the Sulforhodamine B (SRB) assay. A colorimetric assay consisted of staining the cellular protein content of adherent cells. Cell lines were seeded in 96-well plates at a density of 20,000 cells per well at 37 °C with 5% CO_2_. They were then incubated for 24 h to allow their adherence, and subsequently, they were treated with different concentrations of genistein (3.7–740 µM), as well as with the vehicle used (Dimethyl sulfoxide DMSO at 1%) as growth control. After each treatment for 24 and 48 h, cells were fixed with trichloroacetic acid (PanReac AppliChem, Barcelona, Spain) for one hour at 4 °C. Cellular proteins were determined by staining with 0.4% SRB (Sigma-Aldrich) for 30 min at room temperature. Subsequently, 5 washes with 1% acetic acid were performed. For these latter procedures, the plate was allowed to dry at room temperature. SRB-bound proteins were solubilized with 10 mM Tris-base and the reading was performed by absorbance at 490 nm in a microplate reader (Bio-Rad iMark^TM^, Hercules, CA, USA).

### 3.7. Antiproliferation Assay

The antiproliferative effect of genistein was also evaluated with SRB. Cell lines were seeded in 96-well plates with a density of 2500 cells per well at 37 °C with 5% CO_2_. The cells were incubated for 24 h to allow their adherence and were subsequently treated with the vehicle as growth control (1% DMSO) and with 5 different concentrations of genistein based on the IC50 found in the cell viability assay (3.7 µM, 18.5 µM, 37 µM, 185 µM, and 370 µM). The evaluation times were 0, 2, 4, 6, and 8 days. Every 48 h, the culture medium was changed with the respective treatments [29]. After completing each evaluation time, cells were fixed, stained, washed, and read in the same way as in the cell viability assay. For all assays, quintuplicate evaluations were performed.

### 3.8. Reactive Oxygen Species (ROS) Measurement Assay

For the determination of reactive oxygen species, a quantitative analysis was carried out in the Varioskan Lux reader (Thermo Fisher Scientific, Waltham, MA, USA) by measuring fluorescence in a 96-well plate using fluorescein 2′, 7′ Dichlorodihydrofluorescein Diacetate (DCFH-DA) (Calbiochem, San Diego, CA, USA) that allows ROS detection. In this procedure, cells were incubated in 6-well plates at a density of 250,000 cells per well for the SW480 cell line and 350,000 cells per well for the SW620 cell line at 37 °C with 5% CO_2_. Cells were incubated for 24 h to allow their adherence and were subsequently treated with the vehicle as growth control, with the positive control (ferrous sulfate) and the IC50 of genistein corresponding to each cell line and evaluation time. The treatments were evaluated for 24 h and 48 h. After this time, washings were carried out with DMEM medium, DCFH-DA staining, and cell lysis and reading in the varioskan equipment at an excitation wavelength of 485 nm and an emission wavelength of 530 nm. Fluorescence intensity was normalized from the total protein concentration measured for each treatment. Triplicate evaluations were performed for this assay.

### 3.9. Apoptosis Assay

The detection of DNA fragmentation related to cell death processes was detected by flow cytometry using the APO-DIRECT^TM^ kit (Chemicon^R^ International, Temecula, CA, USA) allowing a staining method that marks DNA breaks. In this procedure, cells were incubated in T75 flasks at a density of 1.1 million cells per flask for the SW480 cell line and 1.6 million cells for the SW620 cell line at 37 °C with 5% CO_2_. Cells were incubated for 24 h to allow their adherence and were subsequently treated with genistein (IC50 of each cell line) for 24 h and 48 h. After this time, cells underwent cell fixation, the addition of staining solution, incubation, and subsequent reading in the flow cytometer (LSR Fortessa; BD Biosciences, San Jose, CA, USA), according to the kit instructions.

### 3.10. Determination of Immunological Markers and Apoptosis

For these evaluations, the Magpix equipment was used (Luminex XMAP, Austin, TX, USA), which allows a qualitative and quantitative analysis of proteins. Cell lines were seeded in 6-well plates at a density of 250,000 cells per well for the SW480 cell line and 350,000 cells per well for the SW620 cell line, incubating at 37 °C with 5% CO_2_. Cells were incubated for 24 h to allow their adherence. Subsequently, they were treated with the vehicle as growth control (DMSO at the highest concentration used with the compound in the test) and with the concentrations of genistein corresponding to the IC50 of 24 h and 48 h, evaluating these same times. For the immunological marker measurement assay, the supernatant was collected at each of the evaluated times and taken to the equipment for measurement following the protocol indicated by the manufacturer; the kit used in this assay was the Th1/Th2/Th9/Th17 Cytokine 18-Plex Human ProcartaPlex™ Panel. For the apoptosis assay, a lysis buffer was used to release intracellular proteins and was subsequently taken to the equipment for measurement following the protocol indicated by the manufacturer. The kit used in this assay was Apoptosis 6-Plex Human ProcartaPlex™ Panel (Invitrogen, Waltham, MA, USA). Triplicate evaluations were performed for these assays. The samples were normalized in protein concentration for analysis.

### 3.11. Statistical Analysis

In the cell viability assays, non-linear regression was applied to find the IC50. The results were expressed as the mean ± standard deviation (SD). For the antiproliferation, apoptosis, ROS, and expression of immunological markers assays, a two-way ANOVA was applied followed by the Sidak test after verifying the assumption of normality of the data using the Kolmogorov–Smirnov test. Statistical analyses and graphs were performed using the GraphPad Prism Version 8 program. In all cases, a value of *p* < 0.05 was considered significant.

## 4. Discussion

Many studies show that genistein has great potential as a chemopreventive agent for different types of cancer [10,14,15,30,31]. However, when this compound is administered orally, it has very low stability and bioavailability, due to its insolubility in water [11,32]. Therefore, its encapsulation could improve not only its bioavailability, but also its stability (against oxidizing agents) and its efficacy at the target site [33]. The choice of a good encapsulating agent will allow the compounds to move through the gastrointestinal system, without changes or loss of activity, and to be released at the specific site where the action of this chemopreventive agent is required. BNC is a biopolymer that can be obtained in large quantities through the fermentation of obligate aerobic bacteria of the genus *Komagataeibacter* [34]. In the food industry, BNC is known as nata de coco (Okiyama et al., 1993), and in 1992 it was accepted by the US Food and Drug Administration (FDA) as generally recognized as safe (GRAS) [35]. As an encapsulating agent, BNC allows the incorporation of active ingredients and allows their controlled release [36,37]. BNC has a high aspect ratio in its crystalline nanofibers [38], and when dried by the spray method, the nanofiber network collapses in response to water evaporation and the establishment of irreversible hydrogen bonds [36,37]. As observed in the micrographs, this type of structure contributed significantly to the effective trapping of genistein, hindering the absorption of liquids and the dissolution of the compound in the stomach, making the release more controlled and prolonged. As observed in Figure 1, the morphology of the capsules obtained by spray drying is irregular and are called crumpled paper, with diameters at the micrometric scale [39].

The release profiles obtained in this research indicate that the BNC/GEN capsules present a release of the compound that depends on the pH of the solution. This behavior is characteristic of NCB-encapsulated and oven-dried compounds [40]. The low percentage of the release of genistein in the stomach fluid indicates that BNC protects the compound from the acidic conditions of the gastric fluid [33]. For its part, the increased release of genistein in colonic fluid is attributed to the pH of the solution (pH = 7.4), which alters the hydrogen bonds of the BNC nanofibers, allowing it to open or relax its structure, and altering the hydrophobic interactions between genistein and BNC that allow the release of the compound [33,36]. Finally, the curve of the release profile of BNC capsules loaded with genistein in colonic fluid is consistent with a controlled or prolonged release of drugs since it maintains an almost constant rate from 15 h to 72 h. This allows the concentration of the compound to be maintained in the therapeutic window and to be below the toxic level and above the subtherapeutic level [41]. The results obtained show that BNC acts as a protective agent for genistein in the stomach and allows its release and absorption in the small intestine and colon, where the enterocytes would be responsible for metabolizing this compound [42].

Although genistein has been described as an agent with anticancer properties in several types of cancer, including CRC, there are many mechanisms that need to be described in relation to its biological activity [30]. The measurement of cell viability plays a fundamental role in cell culture assays. The results showed that genistein inhibits cell growth in a dose-dependent manner in the SW480 and SW620 colorectal cancer cell lines, as reported in other studies [16,43]. There are also several studies that have shown that treatment with genistein inhibits the growth of other types of colon adenocarcinoma cells, but in general, all these investigations lack the analysis of the compound in healthy cells that would allow one to define the selectivity of genistein [14,15,43]. In our work, and based on the IC_50_ of each cell line after treatment with genistein, it was possible to determine a selectivity of 3.17 for the SW480 cell line at 24 h of evaluation and 1.75 at 48 h of evaluation. For the SW620 cell line, at 24 h the selectivity was 1.34 and at 48 h it was 1.47 against the non-malignant HaCaT cell line. Additionally, to date, the evaluation of the effect of this type of compound on a tumor cell line and its metastatic derivative, both from the same patient, had not been reported in the same study. Based on these IS and IC_50_s, an antiproliferative effect was observed over the time points tested in a dose-dependent manner for both the SW480 cell line and the SW620 cell line. This inhibition of cell growth is associated with the cumulative dose of genistein that was given every 48 h for up to eight days.

One of the mechanisms by which a compound can inhibit cell growth and proliferation is by inducing apoptosis in malignant cells. In cancer cells, the mechanism of programmed cell death is reduced, demonstrating an imbalance in the proteins involved in the apoptosis process, both proapoptotic and antiapoptotic [44,45]. In this study, an increase in the population in the process of cell death, evaluated by flow cytometry, was observed both at 24 h and 48 h after treatment with genistein in the SW620 cell line. This result agrees with that observed during the measurement of proapoptotic proteins, where a significant increase in the levels of these proteins evaluated by Magpix was observed after treatment with genistein for 24 h and 48 h in this same cell line. After 24 h of treatment with genistein, an increase in Caspase 3 and cleaved PARP proteins were found. Caspase 3 is an essential part of the apoptosis execution pathway. During carcinogenesis, it is deregulated and can be used as an indicator in the progression of the disease. Studies have been described where low levels of caspase 3 indicate decreased apoptosis during tumorigenesis and could be significant in disease progression [15,46]. Caspase 3 is also known to activate other important proteins in this process, including cleaved PARP, a protein that plays several biological roles such as DNA repair and cell cycle regulation. Following DNA damage, a rapid signaling cascade is generated at injury sites to activate cell cycle checkpoints and/or apoptosis to ensure efficient DNA repair. If not repaired, this damage results in an abnormal cell cycle. PARP cleavage occurs by caspase 3, as well as caspase 7, downstream of this signaling cascade to continue the process of apoptosis through the degradation of nuclear material and subsequent cell death [47,48]. PARP cleavage by caspases has also been described as a marker of apoptosis in several studies, specifically in colon cancer cell lines [49]. After 48 h of treatment with genistein, in addition to the proteins mentioned above, a significant increase was found in the proapoptotic proteins p53 and Cytochrome c, and no significant differences were found in the levels of the antiapoptotic protein BCL2, compared to the control. The p53 protein has various functions; among them, it acts as a tumor suppressor, participates in negative regulation of the cell cycle by inhibiting cell division, and has proapoptotic activity depending on the physiological circumstances and the cell type. In the induction of apoptosis, the intrinsic signaling pathway is induced by the p53 protein. This is mediated by the stimulation of the expression of BAX and FAS, as well as by the repression of BCL2 [50]. The p53 protein is also responsible for mediating apoptosis directly in the mitochondria, and after various interactions, it allows the release of Cytochrome c, which regulates the supply of cellular energy. Under conditions of cellular stress, the release of Cytochrome c from the mitochondria is an important step for apoptosis, leading to apoptosome formation, caspase activation, and cell death. The suppression of antiapoptotic proteins, such as BCL2, or the activation of proapoptotic proteins also belonging to the BCL2 family lead to an alteration in the permeability of the mitochondrial membrane that results in the release of Cytochrome c in the cytosol, binding to the Apoptosis Activating Factor 1 (Apaf-1) and triggering the activation of Caspase 9, which then accelerates apoptosis by activating other caspases, such as caspase 3, as previously described [51,52]. The above-described process allows us to identify, in the SW620 cell line, that possibly the apoptosis process observed with genistein treatment occurs through the permeabilization of the mitochondrial membrane in which reactive oxygen species (ROS) may be involved, which were also increased, suggesting an activation of the intrinsic pathway of apoptosis.

ROS are highly reactive molecules and their generation in cells occurs in a balanced manner. They have various functions in normal physiological regulation, such as cell-cycle progression, proliferation, differentiation, and cell death. Additionally, they play an important role in the activation of various cell signaling pathways, and their high levels can lead to damage to proteins, nucleic acids, lipids, membranes, and organelles, leading cells to a process of apoptosis [53]. Higher than normal levels of ROS are typically found in cancer cell lines, helping to promote cancer development and progression. However, some anticancer therapeutics have been described, inducing apoptosis by further increasing cancer cells without affecting non-malignant cells [54]. In this work, after treatment with genistein, an increase in ROS production was observed in the SW620 cell line both at 24 h and 48 h and in the SW480 cell line only at 24 h. There may be a relationship between the production of ROS and the apoptotic process described in the SW620 cell line at both times of treatment with genistein, since an increase in ROS can induce the activation of the ASK1/JNK signaling pathway, which precisely allows the activation of ASK1 sending signals for the activation of JNK and its activation-inducing apoptosis through the mitochondrial signaling pathway, leading to the release of Cytochrome c and continuing the apoptosis process by activating caspases or, also, by activating pro-apoptotic genes [55].

In the SW480 cell line, no levels of these proteins were detected at any of the evaluation times after treatment with genistein. However, a population in the process of cell death was observed in the flow cytometric assay. This test allows us to identify ruptures presented in the process of DNA fragmentation, a process that not only occurs in apoptosis but also in other mechanisms of cell death, including necroptosis, which can also be evaluated and identified with Propidium Iodide (PI) and presents a signaling pathway independent of the caspase pathway and the proteins involved in the apoptosis process [56]. Therefore, it is possible that the cell death process observed for the SW480 line could occur through necroptosis, a non-apoptotic mechanism of cell death that has similarities to necrosis in relation to its morphological characteristics such as loss of membrane integrity and damage to intracellular organelles, as necrosis is an unregulated and unprogrammed cell death where there is a rupture of the cell membrane, the release of intracellular components, and therefore inflammation in adjacent tissues, while although in necroptosis the release of intracellular components is similar to that presented in necrotic cells, the mechanisms are different and occur in a regulated manner. For this reason, necroptosis is described as a form of regulated necrosis or programmed necrosis [56,57,58].

In CRC, cytokines play a crucial role in the development of this type of cancer. However, the available data are insufficient to describe the changes in the cytokine profile during the development of CRC, as well as the mechanisms that lead to changes in the cytokine levels in this type of cancer [59]. In the SW620 cell line, a significant increase was observed, compared to the control, after treatment for 24 h with genistein in the cytokines IL-1beta, IL-2, IL-6, IL-13, IL-17A, IL-27, and GM-CSF. For 48 h of treatment, a significant increase in the cytokines IL-1beta, IL-2, IL-4, IL-5, IL-10, IL-17A, IL-18, IL-27, and GM-CSF was observed. Many of these cytokines have been described with possible antitumor effects; IL-2 antitumor activity appears to be mediated by its effects on NK cells and other cytotoxic cells [60]. It has been used in phase 1 clinical studies combined with other compounds in patients with advanced metastatic colorectal cancer [61]. It has even been used in combination with the drug 5-FU or with other cytokines, comparing its results with chemotherapy used for this type of cancer [60]. As well as this interleukin, although its role is not completely clear in CRC, IL-1beta has also been reported to have antitumor properties. This cytokine has long been associated with inflammation and innate immunity, now describing a broader role that extends beyond classically defined inflammation. Additionally, in the defense of mucosal surfaces, cytokines of the IL-1 family are required, where IL-1beta is found [62,63,64]. IL-5 is critical in the development, activation, and survival of eosinophils, which have been associated with an antitumor response in CRC [65]. Its presence in tumors can influence the activation of the immune system and predict a better prognosis. It has been shown mainly with antitumor properties in CRC [59]. Additionally, both IL-5 and GM-CSF could control tumor growth, since the negative regulation of IL-5 and GM-CSF has been shown to increase tumor burden in a murine model of CRC [66]. IL-18, for its part, promotes the antitumor ability of NK cells in colorectal cancer [67]. This has been found to be decreased in tissues from patients with CRC and its low expression has been significantly correlated with tumor size [68]. In relation to IL-27 and CRC, it has been described as a cytokine with antitumor effects, showing not only antiproliferative and antiangiogenic effects by acting directly on cancer cells, but also having indirect antitumor effects driven by immunostimulatory activity in this and other types of cancer [69].

IL-13 has effector functions, including allergic inflammation, tissue remodeling, and fibrosis. It is a structurally and functionally similar cytokine to IL-4, and the components of its receptors are similar. They regulate the immune response and are involved in several neoplasms. Both cytokines are important in the Th2 response. Several studies have shown discrepancies in the antiproliferative effect of these cytokines in various cell lines and CRC clinical studies [70]. These interleukins can have both pro- and antitumor functions, depending on the tumor microenvironment, and among these functions, a possible inhibitory effect on CRC has been described [59].

IL-6 can show both pro- and antineoplastic activity [71] and is responsible for regulating the proliferation of intestinal epithelial cells in relation to its antitumor activity. It presents different mechanisms, such as the promotion of macrophages and the increase in cytotoxic effects of neutrophils on tumor cells. However, the specific mechanism through which IL-6 plays a role during CRC initiation and progression is not completely clear [72] as there is also evidence that there is no increase in this cytokine compared to the control in studies with patients with CRC [73,74,75,76]. IL-10 is also an immunomodulatory cytokine that exhibits both pro- and antitumor characteristics, depending on the tumor microenvironment, showing its behavior in the pathogenesis and progression of CRC [77]. Perhaps one of its main roles is tumor suppression [59]. Associated with this description, IL-10 has been described with antitumor activity in the tumor microenvironment and with neovascularization inhibitory activity in a murine model, fulfilling a role in the suppression of tumor growth [78]. Finally, IL-17A is part of the IL-17 family, where IL-17B, IL-17C, IL-17D, IL-17E (also called IL-25), and IL-17F are also found. For this family of cytokines, a dual role in the development of CRC has been suggested, thus, highlighting the need for new studies related to its effects in this pathology [79]. Several studies have shown that IL-17A did not have statistical significance at different measurement times in patients with CRC, being shown as a cytokine that is not of potential prognosis in patients with CRC since its levels are not increased [73,76,80].

It is important to highlight that the effects exhibited by these cytokines will depend on the tumor microenvironment in which they are found. Perhaps, for this reason, there is no clear immunomodulatory effect or profile due to the fact that they are in vitro tests with single cell lines, an effect that could, perhaps, be presented in a tumor microenvironment to evaluate different processes in relation to the evolution of tumors.

Finally, one of the great problems of chemotherapeutic agents is the resistance that occurs against them. For this and the other reasons previously described, it is of great importance to develop a cancer therapy based on natural products such as genistein. In this sense, the efficiency of drugs used in chemotherapy depends on the mechanisms to exert their action. One of these mechanisms is apoptosis, in which the deregulation of pro- or antiapoptotic genes in tumor cells has been related to increased resistance to chemotherapy [81]. Apoptosis can be induced by the intrinsic or extrinsic pathway. The intrinsic pathway is regulated by the BCL-2 family of proteins, which allows the release of cytochrome c from the mitochondria and interacts with other proapoptotic proteins until reaching cell death. It has been reported that the overexpression of BCL-2 antiapoptotic proteins increases resistance in ovarian cancer cells to cisplatin, paclitaxel, and other chemotherapeutic agents, both in vitro and in vivo [82]. Genistein has been proposed as an agent to combat this resistance since this mechanism has been described in several studies after its administration, and in this work, its effect is proposed through the intrinsic pathway of apoptosis. The tumor microenvironment in drug resistance is also one of the main reasons for relapse during the treatment of various types of cancer. In this microenvironment, the cytokines produced can provide signals for the growth and survival of tumor cells, hence the importance of their measurement in this work [81]. Other resistance mechanisms in cancer occur due to drug inactivation, drug release from cancer cells, repair of cell damage induced by chemotherapy, activation of pathways favorable to survival, or heterogeneity intratumorally that is observed in different types of cancer and that occurs due to several factors at the cellular level that generate genetic variations such as deletions, translocations, or chromosomal rearrangements [83].

In conclusion, genistein has a chemopreventive effect on the colorectal adenocarcinoma cell line SW480 and its metastatic derivative SW620 through cytotoxicity and antiproliferation evaluations. In both cell lines, there is evidence of the expression of immunological markers that plays an important role in the carcinogenic process and metastasis of CRC and that have been described in other studies with antitumor properties in this type of cancer. By exerting a dual role in this process, depending on the tumor microenvironment, they can have various effects on cell lines, in this case, an increase in expression. The production of ROS in both cell lines may be associated with the cell death process through different mechanisms. Thus, due to the lack of response in the evaluated proteins, it is hypothesized that a necroptotic process mediated by ROS production could occur in the SW480 cell line, while in the SW620 cell line, it is suggested that genistein induces the formation of ROS with the consequent activation of intrinsic apoptosis mediated by caspases and p53. Although genistein has great potential as a chemotherapeutic agent, when administered orally, its bioavailability and, therefore, its effect on the target site, in this case the colon, are decreased. Therefore, new strategies such as encapsulation should be used to improve its effectiveness. BNC encapsulation improved the release profile of genistein, protecting it from gastric pH and allowing its release in the colon.

## Figures and Tables

**Figure 1 molecules-27-07042-f001:**
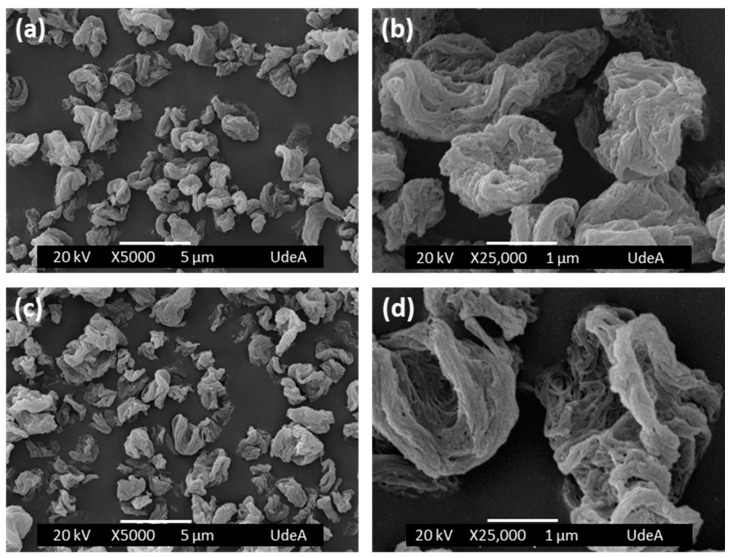
SEM images of spray-dried capsules. (**a**,**b**) BNC capsules and (**c**,**d**) BNC/GEN capsules.

**Figure 2 molecules-27-07042-f002:**
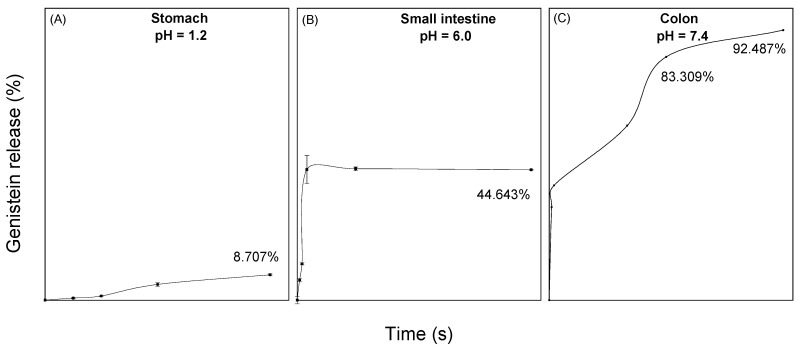
Release curves of genistein encapsulated in BNC at different pH and at 37 °C. (**A**) Simulated stomach fluid at pH 1.2; (**B**) simulated small intestine fluid at pH 6, and (**C**) simulated colon fluid at pH 7.4.

**Figure 3 molecules-27-07042-f003:**
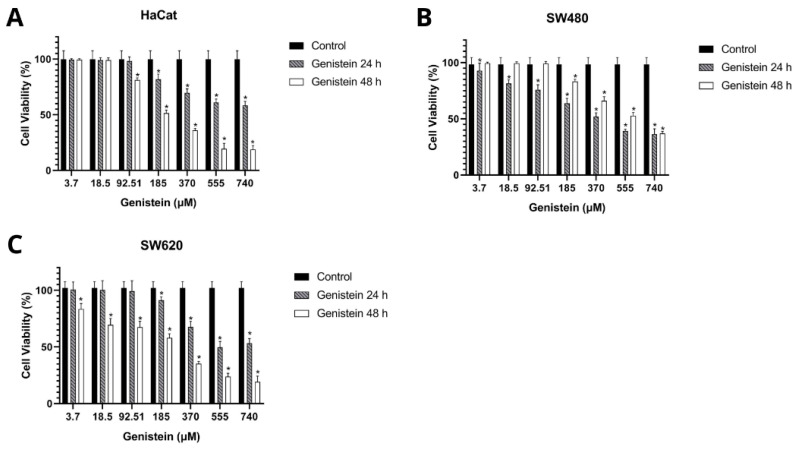
Effect of genistein on cell viability after 24 h and 48 h of treatment compared to the control (**A**) non-malignant HaCaT cell line, (**B**) SW480 colorectal adenocarcinoma cell line, and (**C**) SW620 colorectal adenocarcinoma cell line. All tests were performed in quintuplicate. *p*-value < 0.05 *.

**Figure 4 molecules-27-07042-f004:**
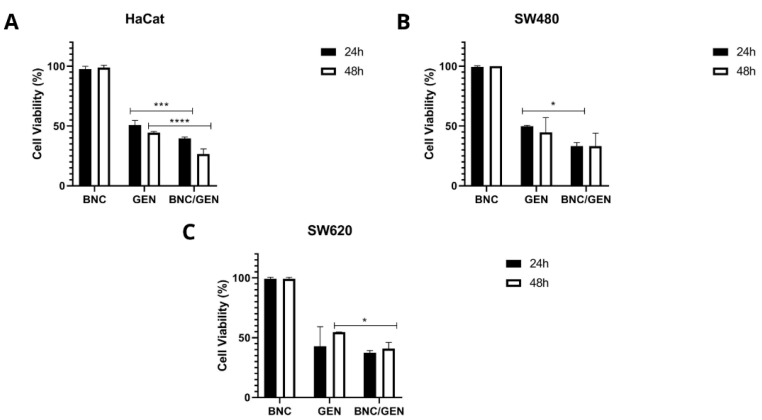
Cell viability of empty BNC capsules and BNC/GEN capsules, taking into account the IC_50_ value of free genistein, after 24 and 48 h (**A**) non-malignant HaCaT cell line, (**B**) SW480 colorectal adenocarcinoma cell line, and (**C**) SW620 colorectal adenocarcinoma cell line. *p*-value < 0.05 *; *p* < 0.001 ***; and *p* < 0.0001 ****.

**Figure 5 molecules-27-07042-f005:**
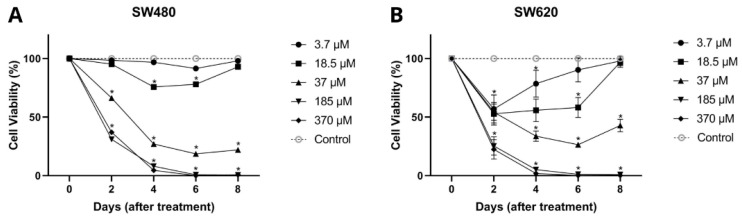
Antiproliferative effect of genistein. (**A**) Colorectal adenocarcinoma cell line SW480. (**B**) Colorectal adenocarcinoma cell line SW620. All tests were performed in quintuplicate. *p*-value < 0.0001 *.

**Figure 6 molecules-27-07042-f006:**
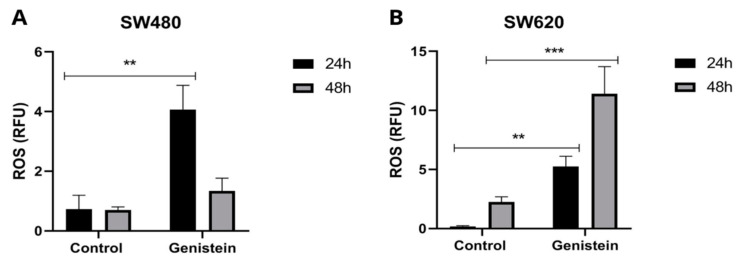
Effect of genistein on the production of reactive oxygen species (ROS) after treatment with genistein in two periods of time (24 h and 48 h). (**A**) SW480 cell line and (**B**) SW620 cell line. All tests were performed in triplicate. RFU: Relative Fluorescence Units. *p*-value < 0.01 **; and *p* < 0.001 ***.

**Figure 7 molecules-27-07042-f007:**
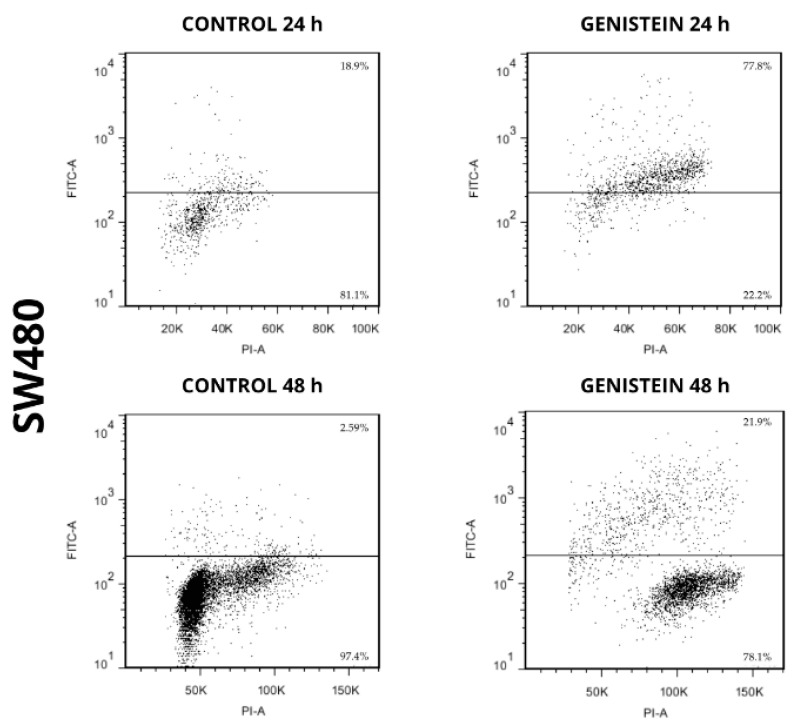
Genistein induces cell death in SW480 cell line. Dot plot of PI/FITC in SW480 cell line after treatment with genistein in two periods of time (24 h and 48 h).

**Figure 8 molecules-27-07042-f008:**
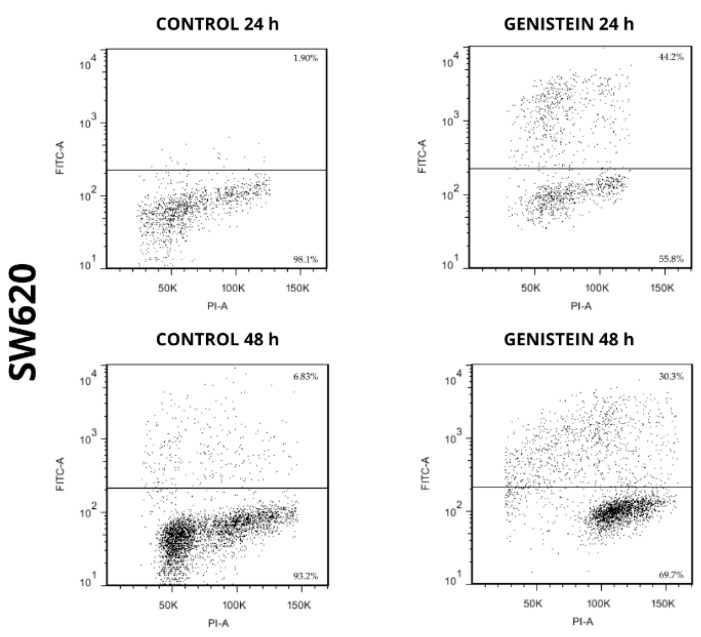
Genistein induces cell death in SW620 cell line. Dot plot of PI/FITC in SW620 cell line after treatment with genistein in two periods of time (24 h and 48 h).

**Figure 9 molecules-27-07042-f009:**
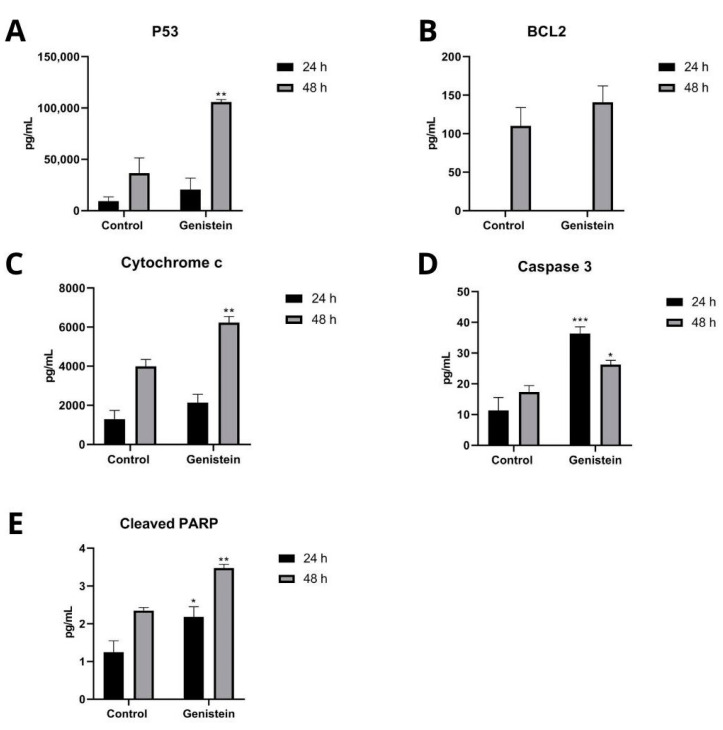
Effect of genistein on intracellular proteins that participate in the apoptosis process in SW620 cell line. Levels of each protein after treatment with genistein in two periods of time (24 h and 48 h) compared to the control. (**A**) p53. (**B**) BCL2. (**C**) Cytochrome c. (**D**) Caspase 3. (**E**) Cleaved PARP. *p* < 0.05 *; *p* < 0.01 **; and *p* < 0.001 ***.

**Figure 10 molecules-27-07042-f010:**
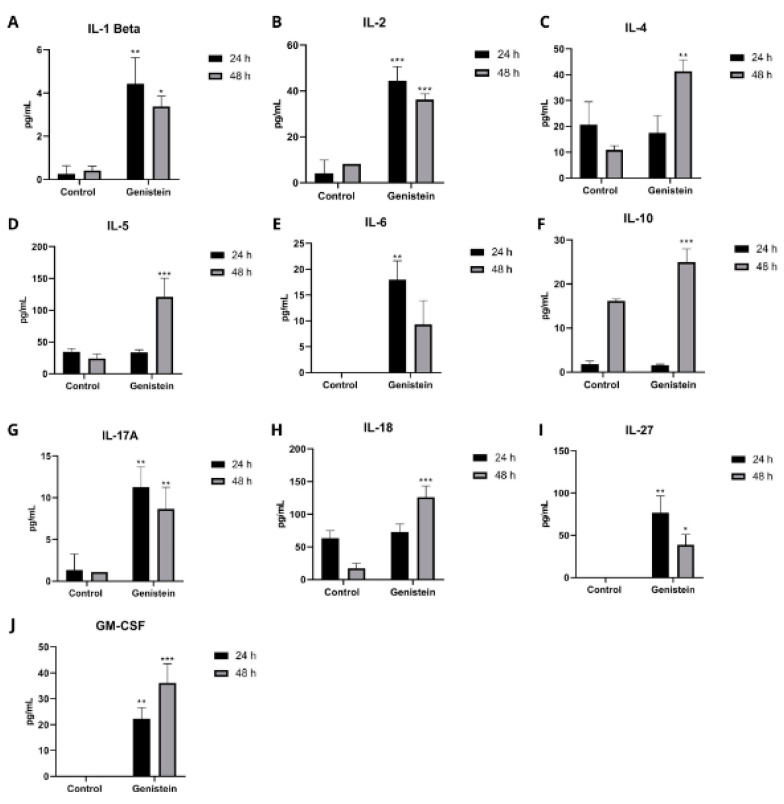
Effect of genistein on important cytokines in the inflammatory and carcinogenic process in SW620 cell line. Levels of each cytokine at 24 and 48 h after treatment with genistein compared to the control. (**A**) IL-1Beta. (**B**) IL-2. (**C**) IL-4. (**D**) IL-5. (**E**) IL-6. (**F**) IL-10. (**G**) IL-17A. (**H**) IL-18. (**I**) IL-27. (**J**) GM-CSF. *p* < 0.05 *; *p* < 0.01 **; and *p* < 0.001 ***.

**Table 1 molecules-27-07042-t001:** Cytotoxic effect of genistein on HaCaT and SW480 cells.

Compound	24 h			48 h		
	IC50 (µM)	IC50 (µM)	SI	IC50 (µM)	IC50 (µM)	SI
	HaCat	SW480		HaCat	SW480	
Genistein	>740	280.93 ± 28.56	3.17	225.01 ± 7.06	128.80 ± 15.87	1.75
5-FU	>962.10	>962.10	2	18.5 ± 3.44	355.23 ± 75.85	0.04

The Inhibitory Concentration 50 (IC_50_) was found using dose–response curves. The selectivity index (SI) was calculated using the formula SI = IC_50_ Non-malignant cells (HaCat)/IC_50_ Tumor cells (SW480).

**Table 2 molecules-27-07042-t002:** Cytotoxic effect of genistein in HaCaT and SW620 cells.

Compound	24 h			48 h		
	IC50 (µM)	IC50 (µM)	SI	IC50 (µM)	IC50 (µM)	SI
	HaCat	SW620		HaCat	SW620	
Genistein	>740	667.17 ± 33.52	1.34	225.01 ± 7.06	152.60 ± 19.13	1.47
5-FU	>962.10	>962.10	0.9	18.5 ± 3.44	139.57 ± 19.09	0.1

The Inhibitory Concentration 50 (IC_50_) was found using dose–response curves. The selectivity index (SI) was calculated using the formula SI = IC_50_ Non-malignant cells (HaCat)/IC_50_ Tumor cells (SW620).

## Data Availability

The data presented in this study are available on request from the corresponding author.

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
