# Peer review of "Evaluation of the Effects of Genistein In Vitro as a Chemopreventive Agent for Colorectal Cancer—Strategy to Improve Its Efficiency When Administered Orally"

_molecules, 2022, doi:10.3390/molecules27207042_

Round 1

Reviewer 1 Report

In this work, Rendon et al shown that genistein display cytotoxic activity on both cancer cell lines SW480 and SW620 while genistein displaying selectivity against non-malignant cell line HaCaT. In addition, the authors shown that Genistein increased production of reactive oxygen species (ROS) suggesting an association with the cell death process through various mechanisms. On other hand, they demonstrate that the encapsulation  with bacterial nanocellulose (BNC) is capable of protecting genistein at acidic pH simulating the gastric fluids.

The article is well written, the experiments are clears and well reported. For the expression of concentration, I suggest to use molarity instead of microgram/mL. I recommend this paper for publication in Molecules

Reviewer 2 Report

Dear Authors:

 I have revised the manuscript: "Evaluation of the Effects of Genistein In Vitro as a Chemopreventive Agent for Colorectal Cancer—Strategy to Improve Its Efficiency When Administered Orally". It is a very interesting manuscript with important results and conclusions. 

I considere it could be accept without revision

Reviewer 3 Report

1) Page 3. The name of the bacterium K. medellinensis must be Italic for the whole of the manuscript. 

2) Fig 2 must be sub-named (A), (B), and (C) since you have 3 figures inside. Why you chose a short release time in the stomach? (120 S, fig 2 left). Moreover, the plot has not reached the plateau. 

3) Cell viability is not the only factor to determine how much a substance prevents cancer cell growth to prove the potential of an agent/drug having an inhibitory effect on angiogenesis or cancer invasion.  Cell toxicity ( by MTT and/or Alamar blue) should be performed. 

4) On page 9, how you calculated cytotoxicity from IC50? You calculated IC50 from the viability test. If any cell does not grow, it does not mean it is dead. It may grow after removing genistein. 

5) Fig 4 is very small. Please arrange them in vertical form and larger with better resolution. 

6) Why the authors did not investigate the pharmacodynamics, pharmacokinetics, and pharmaceutics of the encapsulated genistein? Even if you know the above-mentioned parameters of genistein, for the encapsulated version you have to determine them. 

Reviewer 4 Report

In this manuscript, authros investigated whether the genistein exert antitumor potential on colorectal cancer cells (SW480 and SW620 cell lines). What’s more, author found the protection and release profiles of bacterial nanocellulose on genistein from the acidic conditions of gastric fluid to the colonic fluid. Overall, the authors adopted various measurements, including in vitro release assays, antiproliferation assays, apoptosis assays, and exploration of the expression of cytokines on SW620 cells, suggesting the potential antitumor efficacy of genistein. At this time, there are still some minor problems that authors should carefully address before further consideration.

1. It is better to add the significance of your work that might develop genistein-based cancer therapeutics in the Discussion.

2. Please replace the reference in the antiproliferation assay in Method & Materials section.

4. As Figure 3 shown, the error bars were missing in several groups. Please correct the figure.

6. The number of samples should be presented in the figure legend (Figure 6).

Round 2

Reviewer 3 Report

The manuscript has been improved and the authors answered my comment.